# Osteoarthritis as a Systemic Disease Promoted Prostate Cancer In Vivo and In Vitro

**DOI:** 10.3390/ijms25116014

**Published:** 2024-05-30

**Authors:** Samuel Rosas, Andy Kwok, Joseph Moore, Lihong Shi, Thomas L. Smith, E. Ann Tallant, Bethany A. Kerr, Jeffrey S. Willey

**Affiliations:** 1Department of Orthopedic Surgery, Wake Forest University School of Medicine, 1 Medical Center Boulevard, Winston-Salem, NC 27101, USA; 2Department of Radiation Oncology, Wake Forest University School of Medicine, 1 Medical Center Boulevard, Winston-Salem, NC 27101, USAjwilley@wakehealth.edu (J.S.W.); 3Department of Cancer Biology, Wake Forest University School of Medicine, 1 Medical Center Boulevard, Winston-Salem, NC 27101, USA; 4Department of Hypertension, Wake Forest University School of Medicine, 1 Medical Center Boulevard, Winston-Salem, NC 27101, USA

**Keywords:** osteoarthritis, prostate cancer, COMP, mouse model, destabilization of the medial meniscus

## Abstract

Osteoarthritis (OA) is increasing worldwide, and previous work found that OA increases systemic cartilage oligomeric matrix protein (COMP), which has also been implicated in prostate cancer (PCa). As such, we sought to investigate whether OA augments PCa progression. Cellular proliferation and migration of RM1 murine PCa cells treated with interleukin (IL)-1α, COMP, IL-1α + COMP, or conditioned media from cartilage explants treated with IL-1α (representing OA media) and with inhibitors of COMP were assessed. A validated murine model was used for tumor growth and marker expression analysis. Both proliferation and migration were greater in PCa cells treated with OA media compared to controls (*p* < 0.001), which was not seen with direct application of the stimulants. Migration and proliferation were not negatively affected when OA media was mixed with downstream and COMP inhibitors compared to controls (*p* > 0.05 for all). Mice with OA developed tumors 100% of the time, whereas mice without OA only 83.4% (*p* = 0.478). Tumor weight correlated with OA severity (Pearson correlation = 0.813, *p* = 0.002). Moreover, tumors from mice with OA demonstrated increased Ki-67 expression compared to controls (mean 24.56% vs. 6.91%, *p* = 0.004) but no difference in CD31, PSMA, or COMP expression (*p* > 0.05). OA appears to promote prostate cancer in vitro and in vivo.

## 1. Introduction

Evaluating comorbidities as modifiers of natural disease progression has gained widespread interest [1,2,3,4], which includes considering if/how osteoarthritis (OA) affects the clinical outcomes of patients with prostate cancer (PCa). Importantly, PCa is a highly prevalent disease in the United States, affecting over 54 million adults and causing over 550,000 deaths annually [5,6] with OA also having a large prevalence in the US. For patients with PCa, the 5-year survival rates are favorable (100%) when there is localized or regional disease only, but once the cancer has metastasized, the rate of survival declines to 30% [5]. This last point demonstrates the importance of evaluating predictors of increased metastatic disease in PCa. A recent retrospective study evaluating over 250 patients with PCa demonstrated that those with OA had increased metastatic disease during follow-up, indicating that an association may exist between OA and PCa [7].

New interest in serum markers of OA has pointed in the direction of cartilage oligomeric matrix protein (COMP). This protein serves as an extracellular matrix stabilizer that is upregulated systemically with OA and found continuously increased in high quantities in the serum of patients suffering from this disease. This protein is also implicated in the extracellular matrix organization, and its genetic defects are associated with pseudo-achondroplasia and multiple epiphyseal dysplasia [8]. Interestingly, authors in Sweden found that this protein is also produced by some PCa cell lines [9]. In their studies, the elevated COMP protein concentration was correlated with increased metastatic disease leading to poor survival rates in an immunocompromised mouse model where PCa overexpressed COMP. Englund et al. also identified in 324 tissue samples of PCa patients that tumors expressing COMP were associated with more advanced disease state and increasing invasiveness to regional tissues. Furthermore, this protein has been also found in other types of cancers like colon, liver, and breast cancer [10,11,12,13]. Thus, the increased serum levels of COMP in those with OA led us to believe that it is possible that an association exists between serum COMP and tumor progression. 

Therefore, we hypothesized that a disease that causes an increase in circulating levels of COMP, such as OA might promote PCa progression. Thus, the purpose of our studies was to evaluate whether OA promoted PCa advancement. Our secondary purpose was to determine if COMP would play a determinant role in this relationship. 

## 2. Results

### 2.1. Validation Studies

The RM1 cells did not secrete COMP into the media as seen with undetectable levels of COMP on the ELISA with validated ELISA curves (n = 4, R^2^ = 0.97). Figure 1 demonstrates the significantly greater increase in COMP, MMP-13, and GAGs in the media of cartilage explants treated with IL-1α compared to controls. Moreover, Figure 1 also denotes the pro-inflammatory changes caused by stimulation of explants with IL-1α by demonstrating an increased in P-JNK to total-JNK compared to controls.

### 2.2. Proliferation Studies

There were no significant differences in the growth rates of RM1 cells based on direct application of IL-1α, COMP, or IL-1α + COMP vs. control (*p* = 0.253, Figure 2).

Nonetheless, when evaluating proliferation rates of PCa RM1 cells, a significant difference in the growth rate of the cells treated with OA media was seen when compared to control media (Figure 3). The rate differences were noted after the 10 first hours of the experiment. 

Inhibiting COMP through anti-COMP and COMP downstream pathways with PP2 and Cilengitide did not significantly alter the growth rate of the RM1 cells treated with OA media, with n = 3 experiments with at least two wells per experiment, and *p* = 0.836, thus revealing that COMP is not the sole culprit of increased growth of cancerous RM1 cells stimulated with OA media (Appendix A). PP2 was given at a dose of 2 μM and Cilengitide at 25 μg/mL, as in previous studies.

### 2.3. Migration Studies

Evaluating the effects of direct application of IL-1α and/or with COMP identified no difference in the rate of migration with the scratch wound assay (*p* = 0.650, Appendix A) amongst the studied groups. 

Nonetheless, when migration of cells treated with OA media versus control was evaluated, wounds were closed at a significantly greater rate in wells of cells treated with OA media versus controls (*p* = 0.0158, Figure 4). 

Evaluating whether adding COMP inhibitors (PP2 and Cilengitide), as performed in the proliferation studies, to the OA media altered this relationship, identified that various COMP inhibitors were unable to lead to a significant difference vs. OA media alone (*p* = 0.246, Appendix A). PP2 was given at a dose of 2 μM and Cilengitide at 25 μg/mL, as in previous studies.

### 2.4. In Vivo Results

The mice who underwent DMM successfully developed OA at 8 weeks after DMM surgery as evidenced by an increased OARSI score vs. those without surgery (*p* = 0.041, Figure 5). Tumor implantation was proven in all animals with the 24 h images taken with the IVIS.

ELISAs of COMP collected at the time of euthanasia demonstrated that animals who underwent DMM demonstrated significantly increased levels of COMP vs. control (mean 13.05 pg/mL, SEM 1.51 versus mean 7.06 pg/mL, SEM 0.49, *p* = 0.03). Also, at this time, it was found that 100% of animals with DMM developed a tumor that was able to be seen macroscopically compared to 83.4% in the controls, but this was not statistically significant (*p* = 0.478). Plasma levels of TGF-b were also found to be increased in animals who underwent DMM (mean 7.1 pg/mlt, SEM 0.41 versus mean 5.1 pg/mlt, SEM 0.62, *p* = 0.002).

The OARSI score and the tumor weight at time of death demonstrated a strong positive correlation in animals with OA (Figure 6, Pearson correlation: 0.813, *p* = 0.002 with linear regression). There was no correlation between OARSI score and tumor weight in animals without OA (*p* = 0.583). 

Evaluating tumor growth rates in animals with DMM identified that tumor weight normalized to animal weight at the time of euthanasia did not differ significantly to the non-DMM group (*p* = 0.230, Figure 7), consistent with other reports [9] (*p* = 0.518). The tumor volumes were also not significantly different between the groups (*p* = 0.650, Appendix A).

No significant difference in expression levels of CD31+ vessel coverage (mean 19.04% vs. 27.23%, *p* = 0.485), PSMA (mean 1.97% vs. 2.54%, *p* = 0.541), or COMP (mean 7.10% vs. 6.50%, *p* = 0.342) were identified histologically.

Nonetheless, when evaluating the Ki-67 expression, there was a significant difference in the percentage of cells staining positive with this antibody (mean 24.56% in the DMM group vs. 6.91% in controls; *p* = 0.004, Appendix A). Appendix A demonstrates the expressions levels in different animals. 

## 3. Discussion

The purpose of these studies was to evaluate whether OA would promote PCa progression. We also aimed to evaluate whether COMP was a significant sole contributor to this relationship, as Englund et al. have previously demonstrated that PCas that expressed COMP had worse clinical outcomes [9].

Our in vitro findings demonstrate that OA does stimulate PCa growth at a greater rate than controls. These findings are consistent with our clinical findings [7]. Furthermore, they are likely representative of the additive effects of the multiple cytokines involved in OA. As our findings show, not only is systemic COMP increased with unilateral OA, but so is TGF-b. Also, our findings of increased Ki-67 in the IHC of the PCa tumors of those animals with OA is an interesting finding that did not occur in our retrospective patient studies, but is concordant with an increased amount of COMP expression in the prostate samples [7]. Both of these findings further support our hypothesis that OA can promote PCa by OA representing a systemic inflammatory chronic condition with a multitude of increased serum markers of inflammation including interleukins, adipokines (leptin, adiponectin), and other proteins, including chemokines (COMP, hyaluronan, CCL2) and others, such as TNF-α, various fibroblast growth factors, etc. [14,15].

Of note, many of these systemically increased and circulating factors that are increased with OA have been implicated in a variety of cellular and extracellular processes in prostate cancer [15,16,17,18]. This systemic and progressive inflammatory process in OA likely represents an added chronic stimulus to PCa that can promote the extracellular degradation of surrounding tissue around the primary cancerous process, the growth of the tumor, and/or the migration to end organs.

Our experiments demonstrated that treatment with some of the components of OA such as IL-1α and COMP was not the sole contributing factor causing the cancer to grow and migrate. This is partially explainable, as many of the proteins, cytokines, and growth factors overly expressed with OA are known to stimulate inflammatory pathways known to promote PCa proliferation and migration [19]. 

Of interest, our retrospective studies found that abrogation of the relationship between OA and PCa occurred when arthroplasty was factored into our patient retrospective models [7]. This further confounds the findings of our in vitro experiments with the downstream inhibitors of COMP, as arthroplasty would represent inhibiting the entirety of the inflammatory proteins in patients with OA. This would represent a decrease not only of COMP but also ILs, adipokines, etc. Nonetheless, future studies evaluating the mitigation of OA through multiple pathways may lead to a better understanding of which molecular pathways are critical for the additive effects of OA on PCa. Our hypothesis that COMP or IL-1 was a lead driver of the increased migration and proliferation of PCa was not confirmed but cannot be further excluded in its entirety, as further experiments, including those with continued delivery and at a variety of different conditions and doses, could further evaluate this. Adding a variety of inhibitors to the OA media did not provide significant differences in PCa progression, although our models had inherited limitations. The greatest limitations of these in vitro studies is likely an underestimation of the effects of the inhibitors as they modify the relationship of COMP within the OA media and its effects on PCa. Due to the short horizon of these experiments, it is difficult to assess how two chronic conditions such as PCa and OA present in vitro. In contrast to our findings, Englund et al. found mixed results in regard to how COMP changed PCa proliferation and migration. The authors reported that DU145 cells treated with COMP had decreased growth at 3 days, which was not seen in 22Rv1 cells. Furthermore, they reported that adhesion and migration of DU145 cells were not affected by COMP overexpression. Also, the same authors reported that DU145 cancer invasion into tissues was dependent on the Src and integrin pathway and not on those involving PI3K, JAK-2, NFkB, and RAC, as the cells that were transfected with COMP invaded more than those mock-transfected [9]. The same group’s finding that extracellular COMP also promoted invasion of DU145 PCa cells was reported through more primitive methods based on random pictures of fixed cells, which may have be a cause of the differences in our findings combined with the fact that our cells were RM1 murine Pca, which may behave differently than the human cells used in their studies. RM1 murine PCa features overexpression of the Ras and myc proteins to drive cancer progression and are p53-mutated, while DU145 cells express mutated p53 and are heterozygous for PTEN and the 22Rv1 cells retain both p53 and PTEN expression [20,21,22]. These differences in the mutations driving the different cancer cell lines may affect the expression of COMP. 

In their in vivo studies, Englund et al. also reported that COMP-transfected DU145 cells into mice led to greater tumor volume in their model. Nonetheless, these differences were seen starting at 8 weeks post-cancer injection in their immunocompromised model, which may limit their findings, as COMP is known to also affect cancer cell binding with complement and the various immune pathways thereafter stimulated [9,23]. In contrast, our immunocompetent model could be viewed as more clinically translational and the observation of increased Ki-67 expression in those with OA magnifies our findings that OA may promote cancer growth in a chronic manner. 

Das Roy et al. evaluated the hypothesis of the relationship between autoimmune arthritis and breast cancer metastatic capacity and demonstrated that in their in vivo model, breast cancer metastasized more to bone and lung with concomitant greater levels of vascular endothelial growth factor, tumor necrosis alpha, IL-6, and macrophage colony stimulating factor, which may be similar to our hypothesized theory of increased PCa progression due to systemic inflammation [24]. Similarly, other authors also reported on tissue microarrays from patients that COMP was associated with worse prognosis in breast cancer [11]. 

Nonetheless, the findings seen with our studies suggest that chronic suppressive therapy of OA may be a better pathway to mitigate the progression of PCa than to just treat one of the systemic components of OA. Clinically, this would represent either treating OA early on as to decrease its systemic inflammatory effects or aiming to decrease multiple biomarkers of OA such as COMP or others, known to be chronically overexpressed and that may likely promote disease progression [25,26,27].

A myriad studies, including recent systematic reviews, have aimed to conclude whether different types of arthritis, including OA, promote the progression of various cancers. These meta-analyses have provided various conclusions, but most report limitations in their findings as the studies evaluated are significantly heterogenous [28,29]. Nonetheless, it appears that rheumatoid arthritis has been the main implicated form of arthritis; therefore, we were unable to find epidemiological studies evaluating OA and PCa progression specifically [29,30,31,32]. 

Lastly, our study highlights a novel pathway that may further elucidate mechanisms for cancer promotion in the setting of OA, which could provide a therapeutic treatment strategy by blocking COMP, yet more work is still required.

### Limitations

Our study is not without limitations. As mentioned, in vitro proliferation and migration studies do not replicate clinical states of disease, are short-lived, and may be subject to technical problems, as with any in vitro study. Furthermore, the RM1 cells used in our experiments are not of human origin, and thus, we cannot directly extrapolate our findings to the clinical setting. This cell line was nonetheless chosen as it allowed us to perform the translational model on mice without the need for immunosuppressed mice. Our in vivo study was also limited by the sample size and requirement of subcutaneous injection of PCa. This model was chosen as a tertiary purpose of this study was to examine the feasibility of tracking PCa metastasis with the IVIS. Specific testing on antibody specificity and cross-reactivity between cell types used were also not performed, but previous work has demonstrated that the antibody used is reactive [33]. Future studies should consider the use of syngeneic mouse models such as a transgenic adenocarcinoma of the mouse prostate (TRAMP) model, evaluation of the immune response to the in vivo tumors and those with genetically engineered mice in which PCa develops could add further to the results of our experiments [34,35]. Moreover, studies focusing on tumor microenvironment and in animals without a competent immune system could add in the search for mechanistic explanations to these relationships. These studies would also benefit of evaluation of exosome in the secreted media as these may also ultimately affects the substances releases and their interactions. 

Although these limitations exist, we believe our study was conducted in a manner that would allow other investigators to replicate our studies, and that would provide the best possible translational data. As with clinical studies, other sources of bias may have been introduced inadvertently, yet most experimental analyses were conducted.

## 4. Materials and Methods

### 4.1. In Vitro

#### 4.1.1. Prostate Cancer Cells

The murine RM1 prostate cancer cells were chosen for these studies [36]. They allow for examining their response in vitro as well as in vivo, as they are murine and permit the use of an immunocompetent model. Utilizing the same cell lines in vitro and later in the in vivo mouse model provided a more translational, applicable, and comprehensive model for studying these conditions [37]. The RM1 cells were labeled with green fluorescent protein for ease of tracking on live imaging. The RM1 cells were obtained from a co-author and grown in RPMI supplemented with 10% FBS, penicillin (100 U/mL), and streptomycin (100 µg/mL) in a humidified incubator at 5% CO_2_ at 37 °C. 

#### 4.1.2. Reagents and Antibodies

The following reagents were obtained from the (redacted): Roswell Park Memorial Institute medium (RPMI), fetal bovine serum (FBS), penicillin, streptomycin, trypsin, and ethylenediaminetetraacetic acid (EDTA). The following reagents were purchased: COMP (Abcam, Cambridge, UK), IL-1α (R&D Systems Minneapolis, MN, USA), PP2 (SRC inhibitor, Sigma-Aldrich, St Louis, MO, USA), Cilengitide (α, ß Integrin inhibitor, Sigma-Aldrich, St Louis, MO, USA) and Anti-COMP (Abcam, Cambridge, UK). For IHC the antibodies used were Prostat Specific Membrane Antibody (Dako, Agilent, Santa Clara, CA, USA), Ki-67 (Milipore Sigma, Darmstadt, Germany), COMP (Abcam, Cambridge, United Kingdom) and CD-31 (Abcam, Cambridge, UK).

#### 4.1.3. Proliferation Studies

The RM1 cells were incubated at a density of 4000 cells per well in 96-well plates (Corning, Lowell, MA, USA) overnight to allow them to adhere and grow. The next morning, the 96-well plates were inserted into the IncuCyte ZOOM live cell imager equipped with a Nikon Camera (10× objective). The Incucyte Zoom Software (Essen Bioscience, Ann Arbor, MI, USA) was used to quantify and capture proliferation every 2 h [38,39]. Both regular image data and green phase image channels were used for analysis. Twelve hours after cell seeding, treatment media were completely removed and the cells washed once. Treatments were then delivered in a volume of 200 µlts in each well. Every treatment utilized at least 2 wells (range 2–8) in every experiment. Proliferation experiments were performed 3 times for each test with different cell colonies and averaged. 

Proliferation experiments consisted of direct application of IL-1α at a dose of 10 ngrs/lt as previously reported [40]. COMP was given at a dose of 40 µgrs/mlts and RPMI used as a control [9]. 

Conditioned explant media was obtained by treating chondrocyte explants of healthy, freshly euthanized pigs with IL-1α [41]. Full-thickness chondral explants from the medial and lateral femoral condyles were used. No chondral defects or infections were ever seen in any of the specimens. A 6 mm^2^ biopsy punch was used to create explants of similar size. Then, the explants were set in 12-well plates and allowed to rest in the incubator overnight. The following morning the media was changed and again the explants allowed to rest overnight. The next day, explants were treated with either IL-1α, COMP, IL + 1α + COMP and control (vehicle control of same media as others) at the doses mentioned above. Following 24 h, the media was removed, the explants washed, and fresh media added. Then, the explants remained in that media for 48 h prior to digestion. Explant media were collected in separate experiments. The conditioned media was stored in the 4 °C fridge before use or −80 °C for long-term conservation. Known inhibitors of COMP (Cilengitide, Anti-COMP, and PP2) were also used to further examine our hypotheses [9]. Conditioned media were evaluated for COMP levels to determine the release of COMP following IL-1α. 

#### 4.1.4. Migration Studies

The RM1 cells were used for migration studies and also by utilizing the incubator with the Incucyte device and Incucyte Zoom software (S3, 2018, Sartorius, Göttingen, Germany). The 96-well plates used were the ones recommended by the manufacturer (Essen ImageLock, Ann Arbor, MI, USA). Following the manufacturer’s protocol and following experimentation with optimal cell seeding, 40,000 RM1-GFP labeled cells were added into each well and allowed to adhere overnight [42,43,44]. The following morning, the Incucyte Wound Maker (Essen Bioscience, Ann Arbor, MI, USA) was utilized to perform a scratch wound [45]. Prior to the scratching, the media of the cells were extracted, and cells were scratched with 100 µlts of RPMI in each well (per guidelines). After scratching, the wells were washed, and the treatments added in a total volume of 200 µlts. As in the proliferation, images before the scratch were also taken to evaluate proper homogenous seeding/scratching. Following the scratch, the plates were immediately taken back to the incubator and images taken every 2 h. As above, both direct images and green fluorescent channels were taken. The scratch wound width in microns was used to evaluate migration. Separate experiments were performed for direct application of treatments and treatment with conditioned media. Every experiment was repeated 3 times, with every plate having at least 2 wells (range 2–8) with treatment groups. Following data acquisition, wound width was averaged over treatment groups. Conditioned media were obtained as before, and experiments were also performed with inhibitors. For the latter experiments, 50% of the 200 µlt well was treated with the inhibitors and 50% with conditioned media. 

#### 4.1.5. In Vivo Model

The purpose of this model was to determine whether OA influenced cancer progression in vivo. A total of 48 C57BL/6 male mice, aged 12 weeks (sexual maturity), were used. The destabilization of the medial meniscus (DMM) model was chosen as it is one of the most accurate and translational models of post-traumatic OA available [46,47]. At 12 weeks of age, the DMM procedure was performed. Briefly, following general anesthesia, the menisco-tibial ligament was identified and disrupted. Once the ligament was cut, the menisci were probed as to assess extrusion of the medial meniscus. No gross deformity or cartilage lesions were seen on any animals. Standard layer closure was performed, and the animals were given postoperative analgesics and monitored for complications following the animal care and use facility protocol. Twenty-four animals were utilized to validate our post-traumatic OA model and other 24 animals for the model of the cancer implantation. Animals were housed in groups in standard cages with food and drink ad libitum. Care was taken to evaluate animals daily. The Institutional Animal Care and Use Committee (IACUC) at our institution (redacted) approved these studies. All methods were performed in accordance with the relevant guidelines and regulations. Two blinded reviewers evaluated histology to determine the Osteoarthritis Research Society International (OARSI) score, which is a validated histological outcomes score for severity of OA [48]. 

The 24 animals used for the cancer implantation study consisted of 12 who underwent DMM and at 7 weeks received implantation of cells, and 12 who did not undergo DMM. Being a pilot study, no prior power estimation was performed. Seven weeks following DMM was chosen as a time point for cancer injection as it would guarantee that the animals who underwent surgery would have post-traumatic OA at that point given changes occurring at 4 weeks post-DMM, as described in the literature and as previously seen in other experiments [47]. At 7 weeks post-DMM, 4 × 10^5^ RM1-GFP tagged cells in 100 µlts of media were injected subcutaneously into the left flanks of all (24) the animals [49]. The In Vivo Imaging System (IVIS, Perkin Elmer, Waltham, MA, USA) was used to determine whether the injections were successfully at implanting the cells [49]. Animals were assessed for implantation of the cells within 24 h of injection of cells. Given that this was an immunocompetent model, and tumors were known to grow significantly within a short time period (10–14 days on previous studies), the endpoint chosen was 14 days or ulceration of tumors through the skin. All animals were able to complete the 14-day course, with some being close to ulcerating clinically. This time point would not allow one to evaluate mortality, which was not a primary outcome of these preliminary studies.

At 14 days, following tumor implantation, blood was collected, and euthanasia was performed. The left and right hind limbs were kept, and all the left limbs were fixed in ethanol and paraffin-embedded for sectioning. Right limbs were discarded. Data collection included animal weight, tumor weight, tumor volume as calculated by Englund et al., and gross presence of tumor vs. not [9]. Both knees and tumors were later sectioned at 5 µm thickness slices and oven-baked overnight prior to performing histology. Staining was performed following the manufacturer’s instructions for hematoxylin, eosin and safranin-O on the knees. Prostate-specific membrane antigen (PSMA), CD31 (angiogenesis marker), Ki-67 (proliferation marker), and COMP were used to stain the tumors. Outcomes of interest included tumor weight and volume, OA severity on Osteoarthritis Research Society International (OARSI) score, the correlation between OA grades on OARSI scores, and correlation of tumor characteristics to serum COMP at time of dissection [48]. We obtained TGF-b levels at the time of euthanasia as well to evaluate systemic inflammation. Plasma was utilized due to its availability from these animals. The IHC was evaluated as previously performed regarding the percentage of staining to the region of interest (ROI). A blinded assessor performed the grading and ROI calculations for histology. 

#### 4.1.6. Other Experiments

The ELISA kit testing for COMP, MMP-13, and JNK were used to validate outcomes [50,51]. First, media were tested of where RM1 cells had resided in for 48 h to evaluate spontaneous COMP production by these cells. MMP-13 evaluation was performed on media from cartilage explants to determine OA onset in the explants after treatment with IL-1α. The ELISA kit for total JNK and phospo-JNK was used to further corroborate OA onset on IL-1α-treated cartilage explants that had been digested. The dimethylmethylene blue (DMMB) assay was used to evaluate glycosaminoglycan (GAGs) release and evaluate OA onset as well. 

Statistical analysis was performed based on data type and distribution. For normally distributed continuous data, paired *t*-tests and ANOVAs were used. For non-parametric continuous data, a Mann–Whitney U-test was used. Fisher’s exact tests were used to compare nominal data. Linear regressions were used to evaluate proliferation and migration curves; moreover, the differences in the slopes of the regressions were tested between groups. All tests were performed with SPSS version 20 (IBM Corporation, Armonk, NY, USA) or GraphPad Prism 7.04 (San Diego, CA, USA).

## 5. Conclusions

Osteoarthritis appears to promote PCa progression in vitro and in vivo. Our findings suggest that COMP may likely play a major role in promoting proliferation, yet further studies are needed.

## Figures and Tables

**Figure 1 ijms-25-06014-f001:**
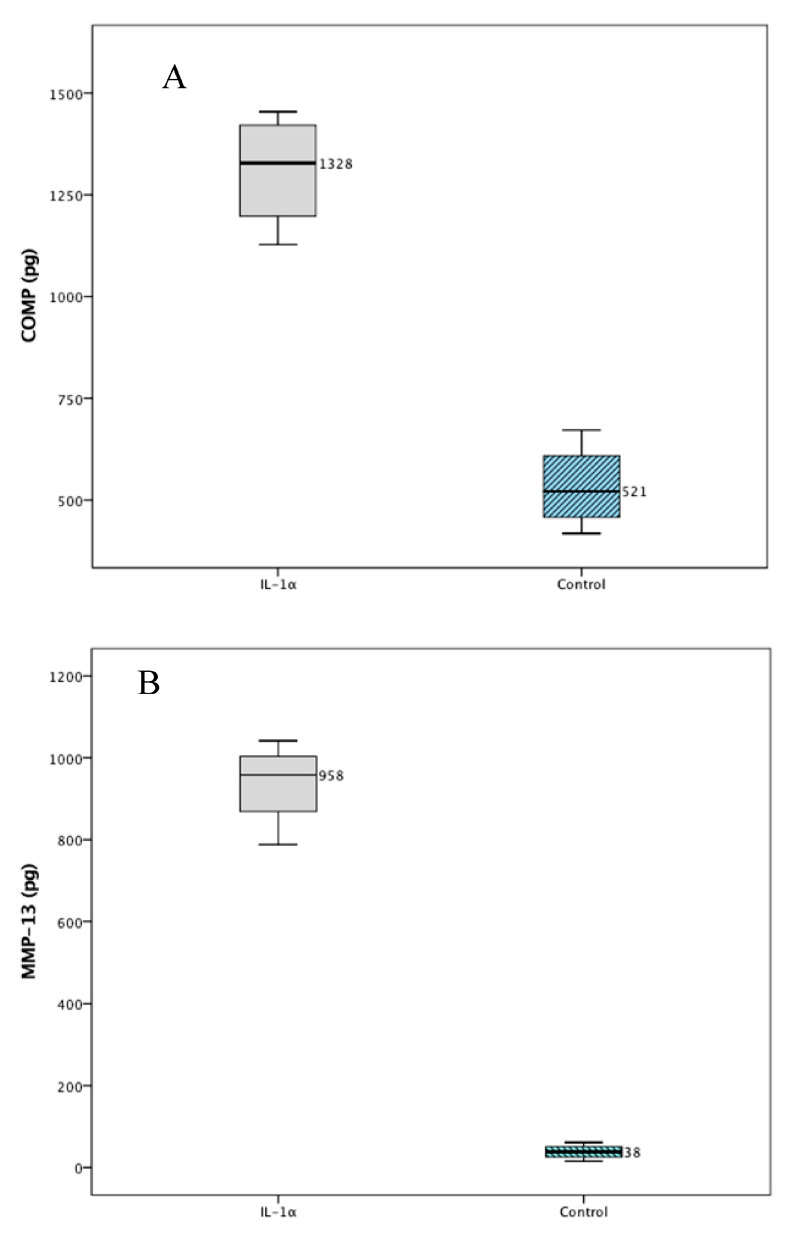
(**A**) COMP media amount 24 h after chondral explant stimulation with IL-1α at 10 pgrs/lt was significantly greater compared to treatment with control media n = 4, *p* < 0.001, paired *t*-test (median, IQR are graphed above). (**B**) MMP-13 secretion into the media was significantly greater for explants treated with IL-1α at 10 pgrs/lt, n = 8, *p* < 0.001, paired *t*-test (median, IQR are graphed above). (**C**) DMMB: There was a significantly greater release of glycosaminoglycan’s into the media of cartilage explants treated with IL-1α10 ugrs/lt for 24 h compared to controls n = 3 experiments, *p* < 0.001, Mann–Whitney U-test (median, IQR are graphed above and outliers represented with circles). (**D**) P-JNK to total JNK ratio was significantly greater in explants treated with IL-1α at 10 ngrs/lt for 24 h compared to controls, n = 5, *p* < 0.001, paired *t*-test (median, IQR were graphed above).

**Figure 2 ijms-25-06014-f002:**
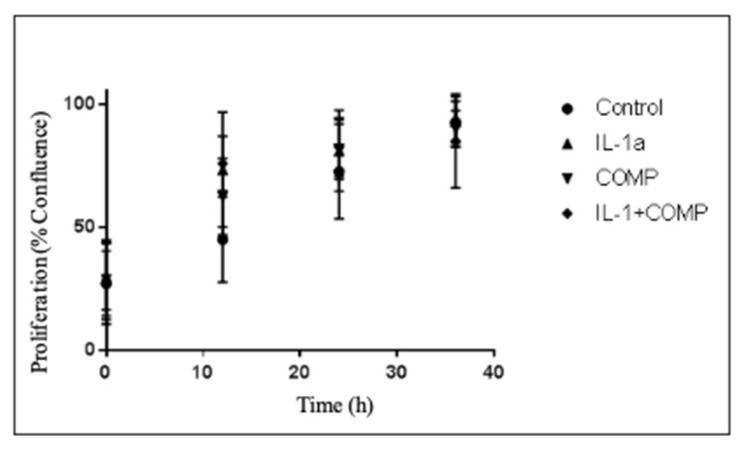
Proliferation over time of RM1 PCa cells based on different components of OA. During the entire timeframe of the experiment, no significant differences were noted between groups. Treatment was performed with IL-1α at 10 pgrs/lt, COMP 40 µgrs/mlts. (*p* = 0.253, linear regression).

**Figure 3 ijms-25-06014-f003:**
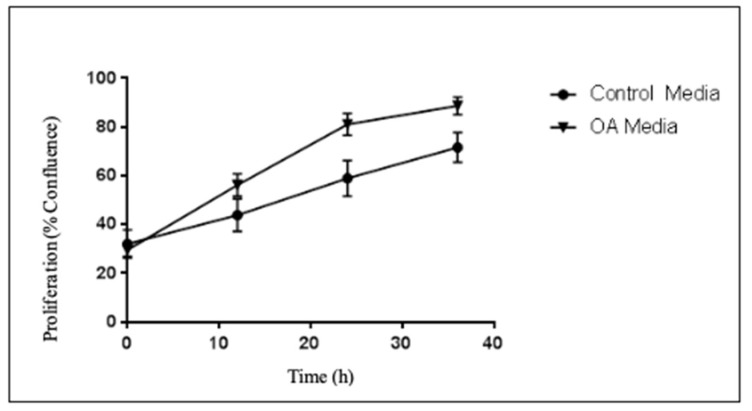
Proliferation over time of RM1 cells based on treatment with control media versus OA media. Experiment performed 3 times with at least 2 wells per treatment group; *p* = 0.0141, linear regression.

**Figure 4 ijms-25-06014-f004:**
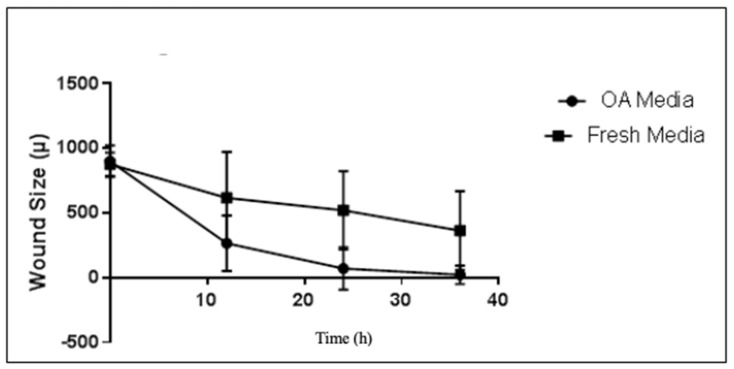
Migration over time of RM1 cells based on treatment with fresh media vs. OA media. RMI cells were incubated with or without OA media and migration was assayed using the IncuCyte wounding assay. n = 3 experiments in duplicate or triplicate (*p* = 0.0158, linear regression).

**Figure 5 ijms-25-06014-f005:**
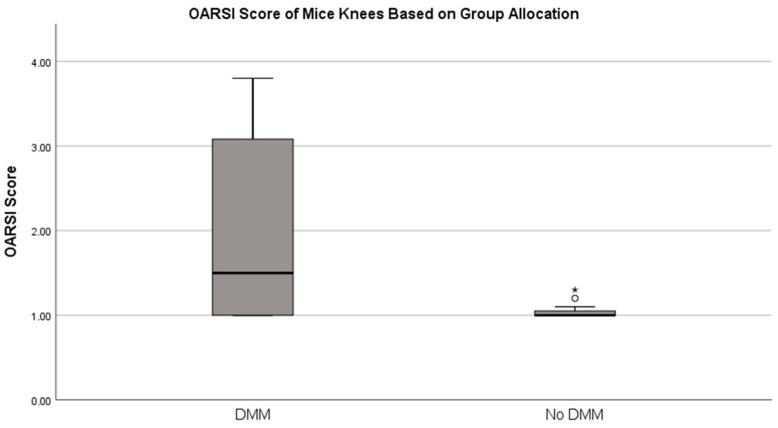
OA severity scores based on OARSI scoring system. OARSI: Osteoarthritis Research Society International. N = 12 per group, *p* < 0.05 * denotes a significant difference between groups (*p* < 0.05, Mann–Whitney U-test) circle represent outlier.

**Figure 6 ijms-25-06014-f006:**
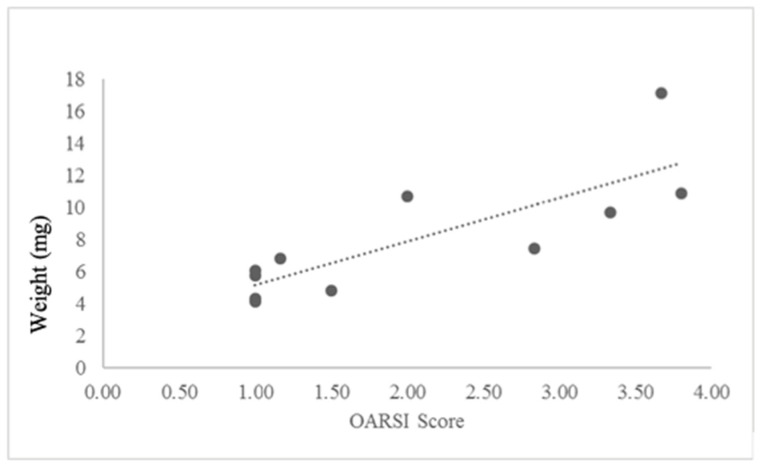
Graphical demonstration of the correlation of tumor weight to severity of OA. The animals in both groups were no different in regard to weight (*p* > 0.05, paired *t*-test), eliminating animal weight as a confounder. Dots represent individual scores, and dashed line represents trend.

**Figure 7 ijms-25-06014-f007:**
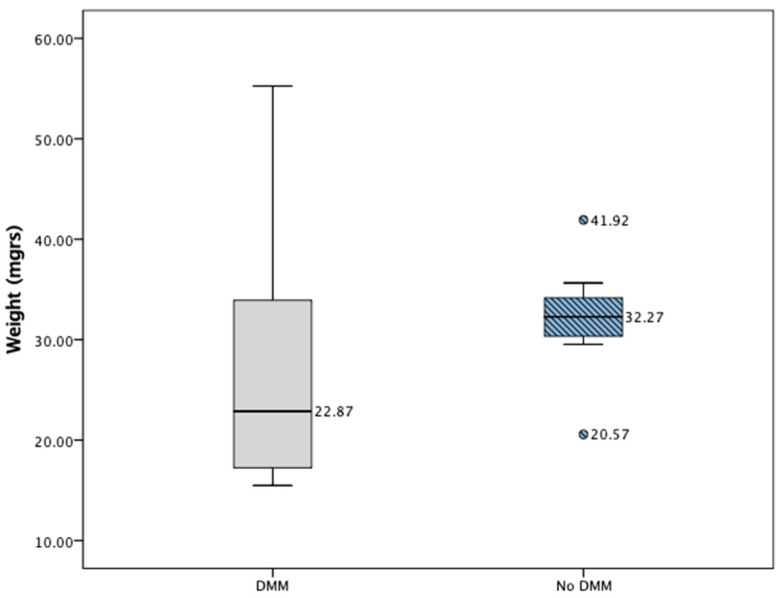
Demonstration of tumor weight according to allocation group (DMM vs. No DMM). *t*-tests demonstrated no significant difference in weight at time of euthanasia (*p* = 0.230).

## Data Availability

Study data are housed within the private network of the Wake Forest University School of Medicine system and can be accessed upon request.

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
