# Peer review of "Osteoarthritis as a Systemic Disease Promoted Prostate Cancer In Vivo and In Vitro"

_ijms, 2024, doi:10.3390/ijms25116014_

Round 1

Reviewer 1 Report

Comments and Suggestions for Authors

Comments:
The manuscript “Osteoarthritis as a Systemic Disease Promoted Prostate Cancer In Vivo and In Vitro” by Samuel Rosa and their colleagues evaluated whether OA promoted PCa advancements and if COMP would play a determinant role in this relationship. Overall, this work is interesting. This manuscript is also well written. However, additional points of clarifications could potentially be addressed to further strengthen the manuscript.

1.     The author has shown the tumor weight results, I was curious if the mice would die (survival rate) of the mice. Does the OA will accelerate the dead speed of the mice?

2.     The author should provide more insights and perspectives on how this study guide readers.

3.     Does COMP only occur in prostate cancer? What about other cancers, dose COMP also related to other cancer? The author should comment on that.

Reviewer 2 Report

Comments and Suggestions for Authors

This article describes experiments conducted using in vitro and in vivo models to investigate the effect of osteoarthritis on prostate cancer progression. In vitro experiments revealed that osteoarthritic cartilage explants promote the proliferation and migration of prostate cancer cells. In vivo studies have shown that osteoarthritis promotes prostate tumor growth and increases Ki-67 expression. The study provides important insights into the link between osteoarthritis and prostate cancer. Although the research is significant, there are two concerns. I recommend that the authors address these points for revision and further discussion in their manuscript.  

The lack of information on the anti-COMP antibody's specificity and cross-reactivity raises concerns about interpreting the results. The use of porcine cartilage explants and murine prostate cancer cells makes it difficult to conclude whether the antibody effectively blocks COMP from pigs.  

Aside from secretory proteins, exosomes in conditioned media can transport a variety of biologically active molecules such as proteins, RNAs, and metabolites, influencing cell behavior. Failure to distinguish between the effects of secreted proteins and exosomes may have missed important biological mechanisms, potentially overestimating the role of secreted proteins alone.
